# FAM111 protease activity undermines cellular fitness and is amplified by gain-of-function mutations in human disease

Saskia Hoffmann[1], Satyakrishna Pentakota[1], Andreas Mund[2], Peter Haahr[1], Fabian Coscia[2], Marta Gallo[1], Matthias Mann[2] (iD), Nicholas MI Taylor[3] (iD) & Niels Mailand[1,4,*] (iD)

## Abstract

Dominant missense mutations in the human serine protease FAM111A underlie perinatally lethal gracile bone dysplasia and Kenny–Caffey syndrome, yet how *FAM111A* mutations lead to disease is not known. We show that FAM111A proteolytic activity suppresses DNA replication and transcription by displacing key effectors of these processes from chromatin, triggering rapid programmed cell death by Caspase-dependent apoptosis to potently undermine cell viability. Patient-associated point mutations in FAM111A exacerbate these phenotypes by hyperactivating its intrinsic protease activity. Moreover, FAM111A forms a complex with the uncharacterized homologous serine protease FAM111B, point mutations in which cause a hereditary fibrosing poikiloderma syndrome, and we demonstrate that disease-associated FAM111B mutants display amplified proteolytic activity and phenocopy the cellular impact of deregulated FAM111A catalytic activity. Thus, patient-associated *FAM111A* and *FAM111B* mutations may drive multisystem disorders via a common gain-of-function mechanism that relieves inhibitory constraints on their protease activities to powerfully undermine cellular fitness.

**Keywords** cell fitness; chromatin; DNA replication; human genetic disorders; protease

**Subject Categories** Autophagy & Cell Death; DNA Replication, Recombination & Repair; Molecular Biology of Disease

## Introduction

Proteases regulate virtually all aspects of cell biology. More than 2% of all known human genes encode proteases or protease inhibitors, highlighting the key importance of proteolytic processes in human physiology (Lopez-Otin & Bond, 2008). Deregulation of protease functions impacts multiple cellular pathways and is linked to a range of human diseases such as cancer, arthritis, osteoporosis, neurodegeneration, and cardiovascular disorders (Lopez-Otin & Bond, 2008). Proteases have therefore attracted major interest as potential drug targets and biomarkers (Drag & Salvesen, 2010). However, despite their central importance for cell and organismal fitness, many proteases remain largely unstudied.

Work over the past years revealed that dominant missense mutations in the *FAM111A* and *FAM111B* genes, which encode proteins harboring a C-terminal serine protease domain, are the underlying cause of rare human multisystem syndromes. Point mutations in FAM111A, a putative host restriction factor that has been linked to DNA replication via a PCNA-binding PIP box (Fine *et al*, 2012; Alabert *et al*, 2014; Tarnita *et al*, 2019), give rise to the two phenotypically related disorders gracile bone dysplasia and Kenny–Caffey syndrome, which manifest with a broad spectrum of severe growth abnormalities including thin and brittle bones, dwarfism, facial dysmorphism, and splenic hypoplasia (Unger *et al*, 2013; Isojima *et al*, 2014; Nikkel *et al*, 2014). In the case of gracile bone dysplasia, this is typically accompanied by perinatal lethality (Unger *et al*, 2013). FAM111B encodes a homologous but so far uncharacterized serine protease, mutations in which are causative of a Rothmund–Thomson-like syndrome characterized by poikiloderma, myopathy, pulmonary fibrosis, and tendon contractures (Khumalo *et al*, 2006; Mercier *et al*, 2013, 2015). The range of known patient-associated FAM111A and FAM111B mutations all map to the periphery of their protease domains, yet the molecular basis of how these mutations lead to disease and whether they are caused by loss- or gain-of-function mechanisms is not known. This is compounded by an almost complete lack of insight into the biological roles of these proteases. In particular, the properties and cellular functions of FAM111A and FAM111B enzymatic activities have not been studied. Accordingly, how FAM111 proteases function in normal physiology and how this is undermined by pathological point mutations in these proteins represent significant knowledge gaps.

1  Protein Signaling Program, Novo Nordisk Foundation Center for Protein Research, University of Copenhagen, Copenhagen, Denmark
2  Proteomics Program, Novo Nordisk Foundation Center for Protein Research, University of Copenhagen, Copenhagen, Denmark
3  Protein Structure and Function Program, Novo Nordisk Foundation Center for Protein Research, University of Copenhagen, Copenhagen, Denmark
4  Department of Cellular and Molecular Medicine, Center for Chromosome Stability, University of Copenhagen, Copenhagen, Denmark
   *Corresponding author. Tel: +45 35325023; E-mail: niels.mailand@cpr.ku.dk

In this study, we discovered that elevated FAM111A protease activity represses essential chromatin-associated processes including DNA replication and transcription, triggering rapid cell death by apoptosis. Remarkably, these properties are exacerbated by patient-associated FAM111A mutants, due to the amplification of their intrinsic protease activity. Furthermore, we discovered a physical link between FAM111A and FAM111B and found that disease-associated FAM111B mutants phenocopy the cellular impact of elevated FAM111A proteolytic activity, likewise resulting from a loss of inhibitory constraints on FAM111B catalytic activity. These findings shed first light on the powerful adverse impact of elevated FAM111 protease activity on essential chromatin-associated processes and cellular fitness, and suggest that heterozygous point mutations in human *FAM111A* and *FAM111B* lead to multisystem disease via a common gain-of-function mechanism unleashing their cytotoxic proteolytic activities.

## Results and Discussion

### Human FAM111A and FAM111B are active proteases

The FAM111 family proteases FAM111A and FAM111B harbor C-terminal serine protease domains that have not been mechanistically and functionally characterized (Fig 1A). Sequence analysis of the human FAM111A and FAM111B protease domains revealed that they contain evolutionarily conserved catalytic triads and display homology to stress-responsive *Escherichia coli* Deg-type proteases, suggesting they are catalytically active (Figs 1B, and EV1A and B). Supporting this, *in silico* structural modeling analysis suggested that both the FAM111A and FAM111B active sites adopt conformations with notable similarity to that of the *E. coli* DegS protease (Fig 1C–F). Using purified recombinant full-length human FAM111A and FAM111B proteins, we validated that both are active proteases *in vitro*, as evidenced by the formation of auto-cleavage products that were dependent on an intact catalytic triad and abolished by treatment with the serine protease inhibitor AEBSF (Figs 1G and H, and EV1C–G). Strikingly, we noticed that elevated expression of wild-type (WT) FAM111A but not a catalytically inactive D439N mutant in human cells led to a drastic loss of cell viability, whereas the expression of WT FAM111B had no impact (Figs 1I and 2A; see also Fig 5J). These data suggest that human FAM111A and FAM111B harbor intrinsic proteolytic activity, which at least in the case of FAM111A potently undermines cellular fitness.

### FAM111A proteolytic activity inhibits DNA replication via the RFC complex

The adverse impact of FAM111A proteolytic activity on cell survival prompted us to further explore its cellular function. To this end, we generated stable cell lines conditionally expressing WT GFP-tagged human FAM111A or mutant alleles containing inactivating substitutions in the protease domain (D439N) or PIP box (*PIP) (Alabert *et al*, 2014) at two- to fourfold higher levels than endogenous FAM111A (Fig 2A; Appendix Fig S1A). The higher expression level of the inactive D439N mutant relative to GFP-FAM111A WT likely reflects its lack of cytotoxicity (Fig 1I). Using these cell lines, we assessed the impact of FAM111A

protease activity on DNA replication status, given the known interaction of FAM111A with replication forks (Alabert *et al*, 2014; Wessel *et al*, 2019). EdU incorporation analysis revealed that the expression of ectopic FAM111A in U2OS cells rapidly and potently suppressed DNA replication in a manner that was fully dependent on its protease activity (Fig 2B–D). Elevated FAM111A WT expression also reduced DNA synthesis rates in HCT116 cells (Appendix Fig S1B). Induction of WT but not catalytically inactive FAM111A triggered a substantial decrease in the association of PCNA but not the MCM complex with chromatin in S phase cells (Fig 2C, E and F). The resulting decline in PCNA replication foci intensity was accompanied by RPA accumulation at these sites (Appendix Fig S1C and D), indicating that FAM111A-mediated dissociation of the PCNA module from active replisomes triggers replication stress. While FAM111A has been suggested to promote PCNA loading during DNA replication (Alabert *et al*, 2014), we observed no detectable impact of CRISPR/Cas9- or siRNA-mediated depletion of endogenous FAM111A on PCNA chromatin association, DNA synthesis rates, and cell proliferation (Appendix Fig S2A–H), suggesting it is dispensable for DNA replication *per se* but may have a perturbation-dependent function in regulating this process. Unlike the FAM111A protease domain, mutation of the PIP box that abrogates PCNA binding (Alabert *et al*, 2014) did not impair the ability of FAM111A to suppress DNA replication and PCNA loading (Fig 2A and C–E). This raised the possibility that FAM111A predominantly exerts its protease-dependent impact on DNA replication via other replisome components. Supporting this idea, an unbiased FAM111A interactome analysis revealed the replication factor C (RFC) complex, the main cellular PCNA loader (Shiomi & Nishitani, 2017), but not PCNA itself as a major replication fork-associated cellular FAM111A-binding partner (Fig 2G; Dataset EV1), in agreement with genetic data indicating a link between FAM111A and RFC subunits in restricting viral replication (Panda *et al*, 2017). Co-immunoprecipitation experiments suggested that FAM111A associates with the RFC complex through interaction with the RFC1 subunit independently of the PIP box, involving a region of the extended regulatory N-terminal portion of human RFC1 that encompasses its BRCT domain (Shiomi & Nishitani, 2017) (Figs 2H and I, and EV2A and B). In line with this, endogenous FAM111A and RFC1 colocalized in nucleoli in G1 and G2 phases and formed nuclear foci during S phase (Fig EV2C–E). Ectopic FAM111A expression diminished RFC1 chromatin association but not total cellular RFC abundance in a protease-dependent but PIP box-independent manner (Figs 2J and K, and EV2F), which may account for its suppressive impact on PCNA loading. These findings show that FAM111A proteolytic activity strongly inhibits DNA replication, involving the displacement of both RFC and PCNA from chromatin.

### FAM111A protease activity suppresses transcription and triggers Caspase-dependent apoptosis

We next asked how FAM111A proteolytic activity impairs cell survival. Induction of FAM111A WT or *PIP but not the catalytically inactive D439N mutant in U2OS cells led to robust formation of Caspase 3 and PARP1 cleavage products as well as extensive DNA fragmentation demarcated by accumulation of cells with sub-G1

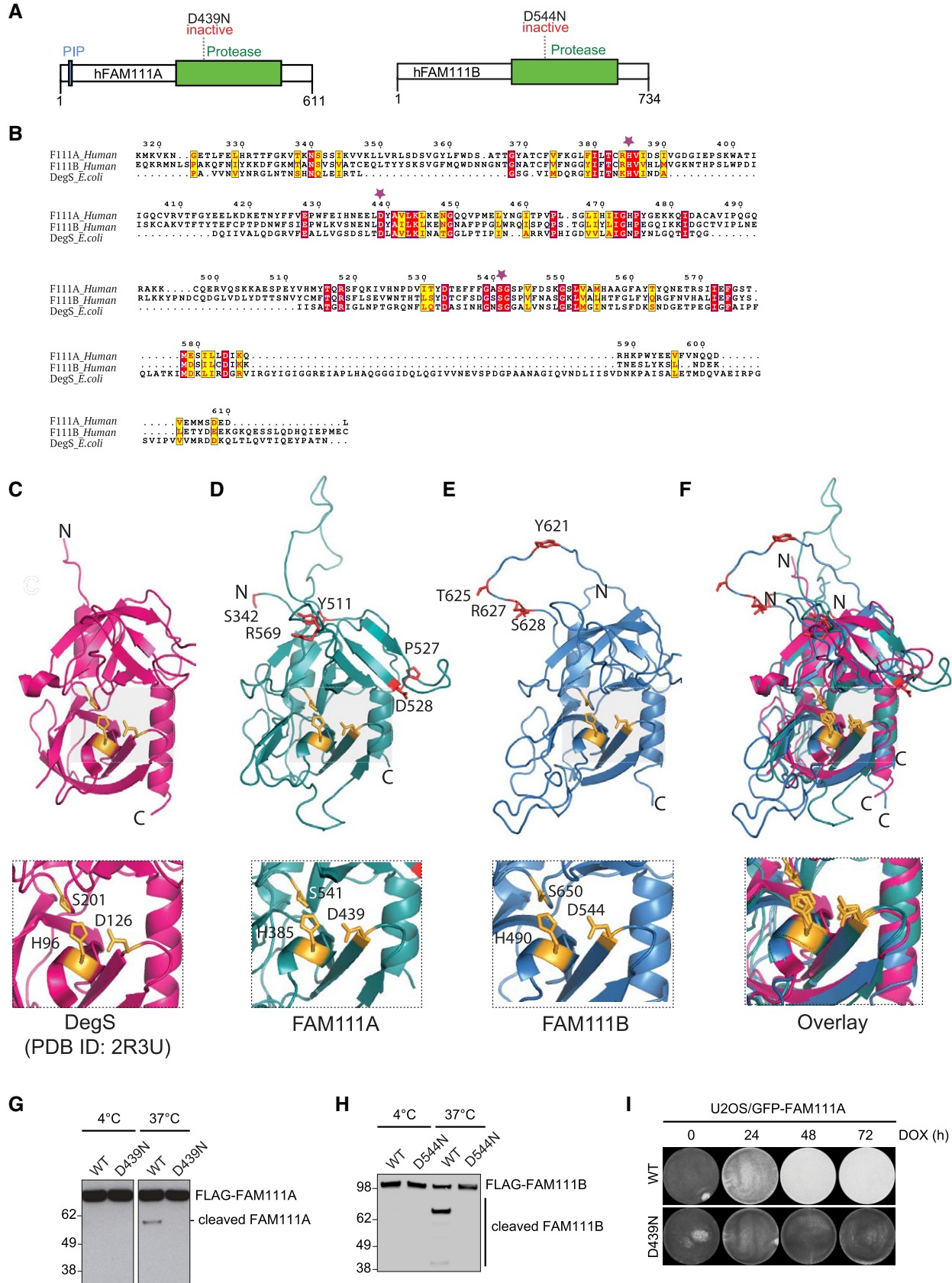

**Figure 1.**

**Figure 1. Human FAM111A and FAM111B are active proteases.**

A   Domain organization of human FAM111A and FAM111B proteins.
B   Alignment of the serine protease domains in human FAM111A, human FAM111B, and *Escherichia coli* DegS. Red boxes denote fully conserved residues; yellow boxes indicate conservative amino acid substitutions; and purple stars indicate catalytic triad residues.
C   Crystal structure of monomeric *E. coli* DegS protease (PDB ID: 2R3U). Zoomed-in view shows position of catalytic triad residues (yellow).
D   Homology-based protease domain model of human FAM111A (residues 371–555; teal). Zoomed-in view shows catalytic triad residues (yellow). Residues mutated in human disease (red) are indicated.
E   Homology-based protease domain model of human FAM111B (residues 471–664; blue). Zoomed-in view shows catalytic triad residues (yellow). Residues mutated in human disease (red) are indicated.
F   Overlay of the DegS, FAM111A, and FAM111B protease domains in (C–E).
G   Purified recombinant FLAG-FAM111A proteins were incubated at indicated temperatures for 4 h, and FAM111A auto-proteolytic activity was analyzed by immunoblotting with FLAG antibody.
H   As in (G), using recombinant human FLAG-FAM111B proteins.
I   U2OS cell lines conditionally expressing indicated GFP-FAM111A alleles were fixed at the indicated times after treatment with doxycycline (DOX) to induce expression of the transgenes and stained with crystal violet. Levels of stably expressed GFP-FAM111A proteins are shown in Fig 2A.

Data information: Data are representative of at least three (G–I) independent experiments with similar outcomes.

DNA content and pan-nuclear γ-H2AX positivity, phenotypes that could be suppressed by the pan-Caspase inhibitor Z-VAD-FMK but were insensitive to p53 status (Figs 3A–D and EV3A–D). We therefore concluded that elevated FAM111A protease activity potently triggers programmed cell death via Caspase-dependent apoptosis. Ectopic FAM111A expression also elicited an apoptotic response in HCT116 cells (Fig EV3E). While FAM111A-mediated apoptosis had no impact on DNA replication suppression and proceeded with slower kinetics, it affected cells at all stages of interphase (Fig EV3F–H), suggesting that the adverse impact of FAM111A protease activity on cell viability is not solely a consequence of disrupting DNA synthesis. Indeed, in addition to its effect on DNA replication, elevated expression of FAM111A suppressed transcription with comparable kinetics in a manner that was largely dependent on its proteolytic activity but neither the PIP box nor cell cycle status (Figs 2B and 3E–G, and EV3I). Consistently, similar to its impact on RFC1, induction of catalytically active FAM111A led to a robust decrease in chromatin-bound but not total levels of RPB1, the catalytic subunit of RNA polymerase II, and FAM111A and RPB1 formed a complex in cells (Figs 3H–J and EV2A). Thus, elevated FAM111A protease activity antagonizes essential chromatin-associated processes including replication and transcription by evicting key effectors of these processes, the collective action of which may lead to rapid cell death by Caspase-dependent apoptosis.

**Patient-associated mutations hyperactivate FAM111A protease activity and its adverse impact on cellular fitness**

Dominant missense mutations in human *FAM111A*, which cluster on the periphery of the protease domain and are predicted by our structural modeling to be surface-exposed (Figs 1E and 4A), are causative of the skeletal disorders Kenny–Caffey syndrome and perinatally lethal gracile bone dysplasia (Unger *et al*, 2013; Isojima *et al*, 2014; Nikkel *et al*, 2014; Abraham *et al*, 2017), yet how these mutations impair normal physiology is not known. To address this, we generated a panel of cell lines conditionally expressing different patient-associated FAM111A alleles or FAM111A WT at a comparable, near-endogenous level (designated WT (low)) (Figs 4B and EV4A–D). Like the stable GFP-FAM111A WT-expressing cell line used above (Fig 2A), the WT (low) cells also suppressed DNA replication albeit with slower kinetics, due to the lower level of ectopic GFP-FAM111A WT expression (Figs 4C and EV4A–C). Strikingly, despite their low abundance, the disease mutants strongly exacerbated the impact of GFP-FAM111A on the kinetics and magnitude of DNA replication and transcription inhibition as well as apoptosis onset (Figs 4B–F and EV4E–I). Consistently, mutant FAM111A markedly enhanced RFC1 and RPB1 dissociation from chromatin (Figs 4G and H, and EV4J and K). We observed similar effects in cells expressing untagged ectopic FAM111A disease mutants, ruling

**Figure 2. FAM111A proteolytic activity displaces RFC from chromatin and inhibits DNA replication.**

A   Immunoblot analysis of stable U2OS cell lines left untreated or incubated with DOX to induce expression of WT or mutant forms of GFP-FAM111A.
B   DNA replication rates in U2OS/GFP-FAM111A cells treated with DOX for the indicated times, pulse-labeled with EdU, and stained with DAPI were analyzed by quantifying EdU signal intensity in S phase cells using quantitative image-based cytometry (QIBC) (red bars, mean (A.U., arbitrary units); n > 2,000 cells per condition). See also Appendix Fig S6A.
C   Cells treated as in (B) were pre-extracted, fixed, and stained with PCNA or MCM2 antibody, and subsequently analyzed by QIBC (n > 2,000 cells per condition).
D–F   Quantification of data in (C) (red bars, mean). Cells in S phase were identified based on EdU positivity. See also Appendix Fig S6B–D.
G   Analysis of FAM111A interactors. U2OS/GFP-FAM111A WT cells were treated or not with DOX for 4 h, subjected to GFP immunoprecipitation (IP), and analyzed by mass spectrometry. Volcano plot shows enrichment of individual proteins (+DOX/−DOX ratio) plotted against the *P* value. Dashed lines indicate the significance thresholds (FDR < 0.05; s₀ = 1).
H   U2OS or U2OS/ΔFAM111A cells were subjected to IP with IgG (control) or RFC1 antibody followed by immunoblotting with indicated antibodies.
I   U2OS cells transfected with empty vector (EV) or indicated RFC subunit expression plasmids were subjected to FLAG IP and immunoblotted with indicated antibodies.
J   As in (C), except that cells were stained with RFC1 antibody (n > 2,000 cells per condition).
K   Quantification of data in (J) for S phase (EdU-positive) cells (red bars, mean).

Data information: Data are representative of at least three (A–F, H–K) and two (G) independent experiments with similar outcomes.

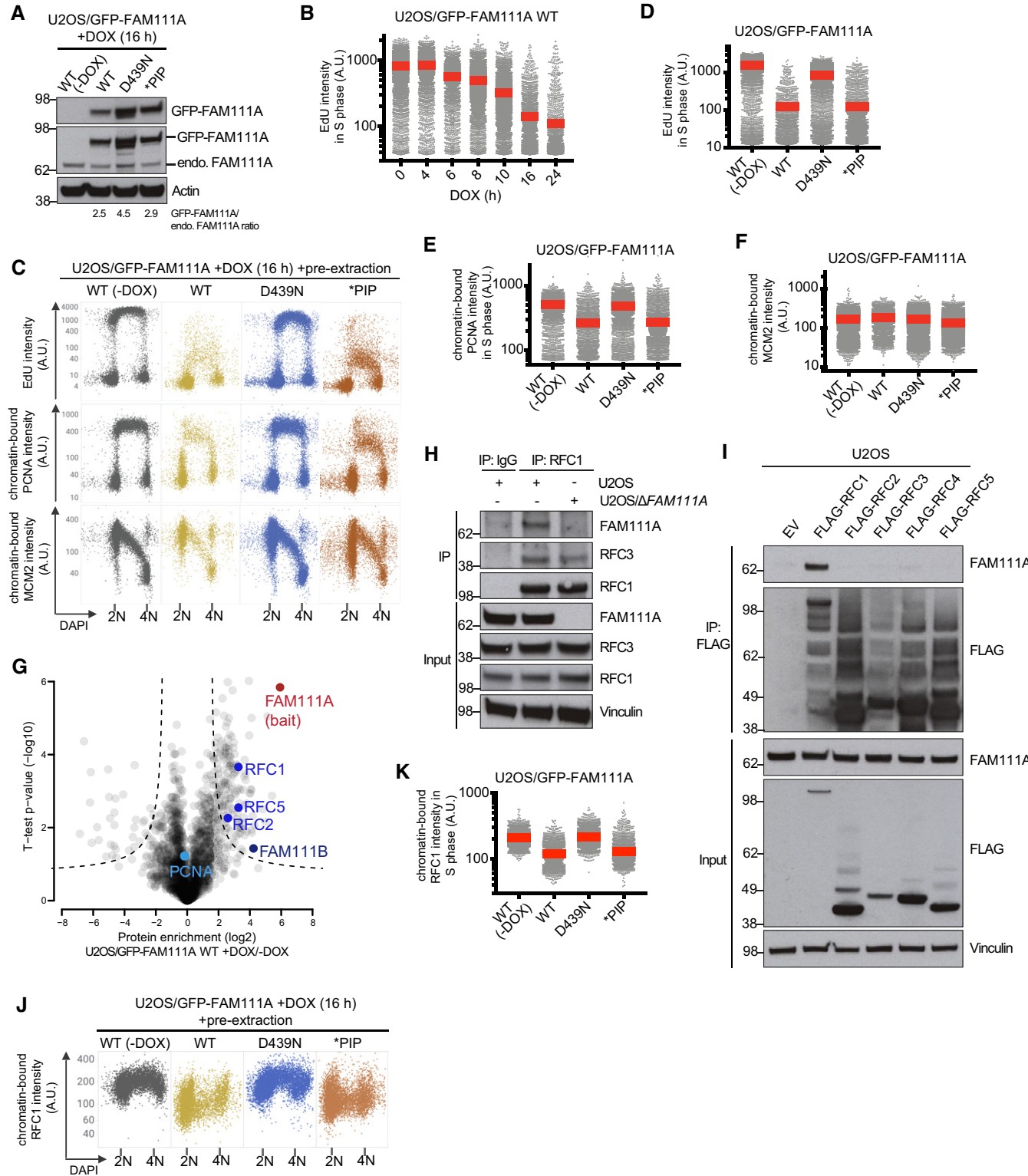

**Figure 2.**

out a contribution of epitope tagging to these phenotypes (Fig EV4L and M). Moreover, complementation experiments showed that sub-endogenous levels of disease-associated FAM111A alleles were sufficient to trigger apoptosis in cells depleted of endogenous FAM111A (Fig EV4M). Introducing an inactivating D439N substitution abrogated the impact of the patient-associated FAM111A mutants on DNA replication, transcription, and apoptosis (Figs 4B–H and EV4I–K), raising the possibility that they aggravate FAM111A-dependent

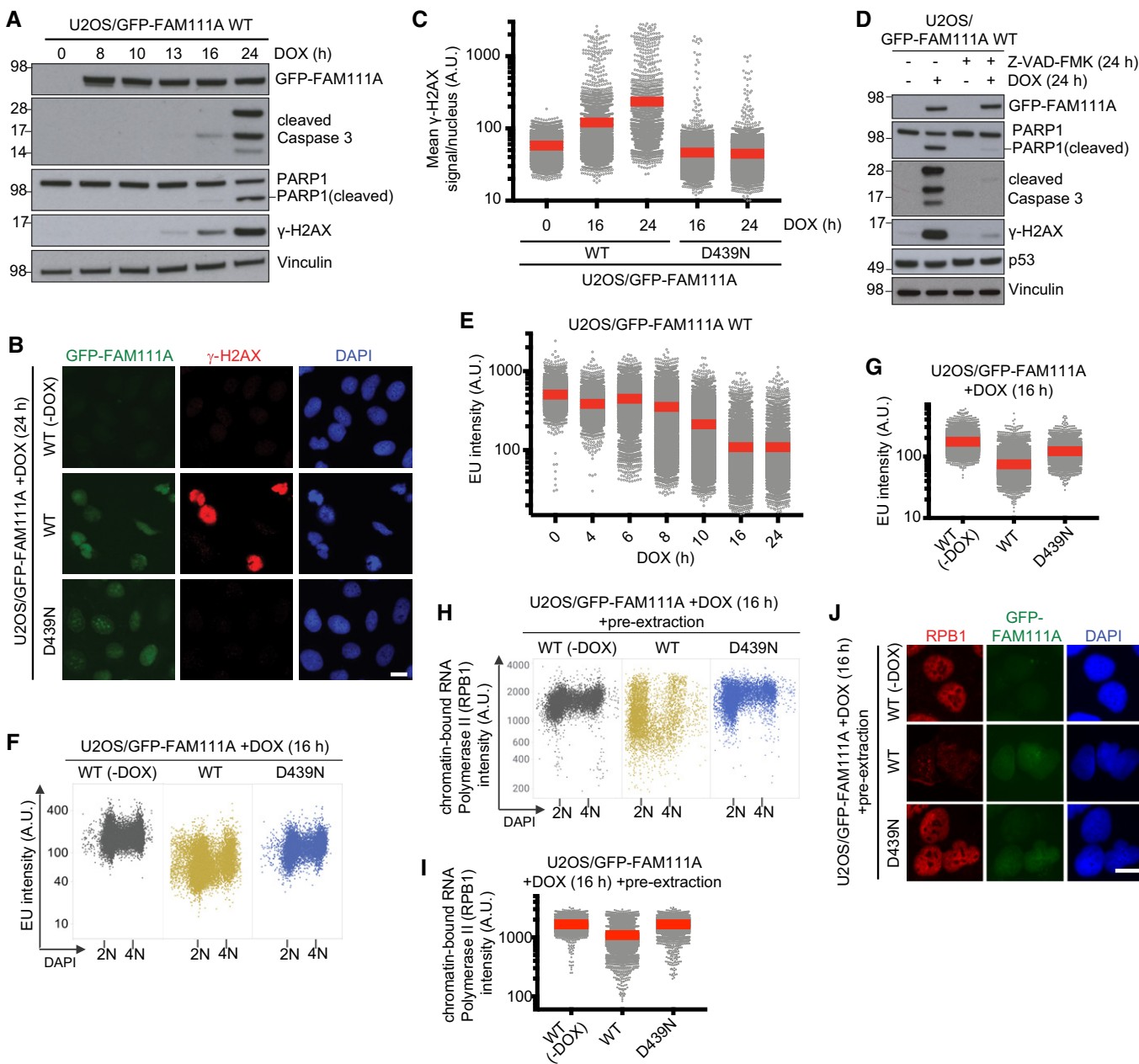

**Figure 3. FAM111A protease activity suppresses transcription and triggers Caspase-dependent apoptosis.**

A    Immunoblot analysis of U2OS/GFP-FAM111A WT cells treated with DOX for the indicated times.

B    Representative images of U2OS/GFP-FAM111A cell lines that were treated or not with DOX, fixed, and stained with γ-H2AX antibody and DAPI.

C    Cells in (B) were subjected to QIBC analysis of γ-H2AX signal intensity (red bars, mean (A.U., arbitrary units); $n > 2{,}000$ cells per condition). See also Appendix Fig S6E.

D    Immunoblot analysis of U2OS/GFP-FAM111A WT cells treated or not with DOX and the pan-Caspase inhibitor Z-VAD-FMK as indicated.

E–G    Transcriptional activity in U2OS/GFP-FAM111A cells treated or not with DOX for the indicated times, pulse-labeled with EU, and stained with DAPI were analyzed by QIBC (red bars, mean; $n > 2{,}000$ cells per condition). See also Fig EV3I for the impact of the FAM111A *PIP mutant and Appendix Fig S6H.

H    U2OS/GFP-FAM111A cell lines treated or not with DOX were pre-extracted, fixed, and stained with RPB1 antibody, and analyzed by QIBC ($n > 2{,}000$ cells per condition).

I    Quantification of data in (H) (red bars, mean). See also Appendix Fig S6I.

J    Representative images from the experiment in (H). Scale bars, 10 μm.

Data information: Data (A–J) are representative of at least three independent experiments with similar outcomes.

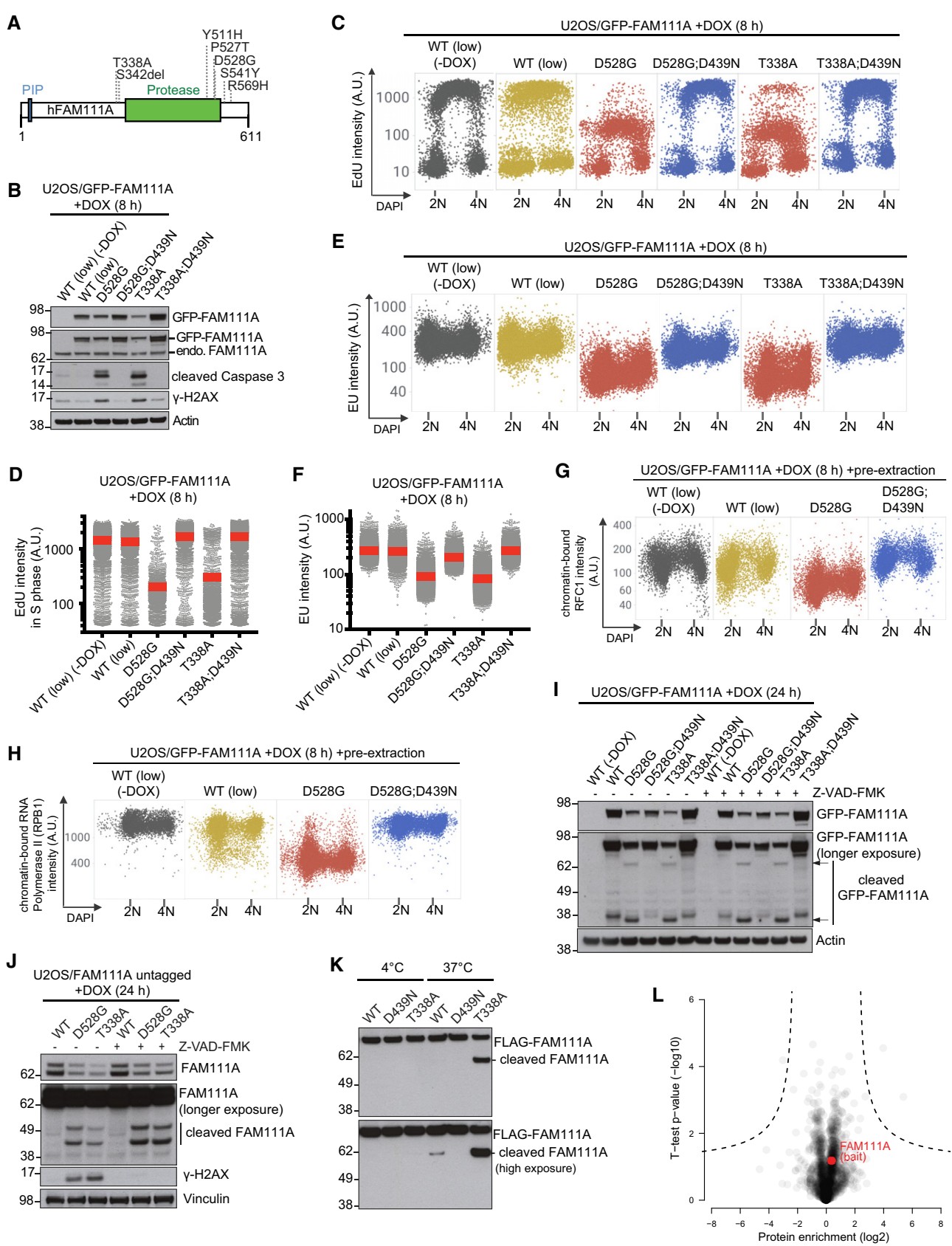

**Figure 4.**

◄

**Figure 4. Patient-associated mutations hyperactivate FAM111A protease activity to exacerbate its adverse impact on cellular fitness.**

A   Overview of heterozygous *FAM111A* mutations found in patients with gracile bone dysplasia or Kenny–Caffey syndrome.
B   Immunoblot analysis of U2OS cell lines left untreated or incubated with DOX to induce expression of the indicated GFP-FAM111A alleles. U2OS/GFP-FAM111A WT (low) cells express the transgene at a lower level than U2OS/GFP-FAM111A WT cells used in Figs 1 and 2 (see Fig EV4A).
C   U2OS/GFP-FAM111A cell lines treated or not with DOX, pulse-labeled with EdU, and stained with DAPI were analyzed for DAPI and EdU signal intensity using QIBC.
D   Quantification of data in (C) for S phase (EdU-positive) cells (red bars, mean (A.U., arbitrary units); $n > 2,000$ cells per condition). See also Appendix Fig S6J.
E   As in (C), except that cells were pulse-labeled with EU.
F   Quantification of EU incorporation in cells in (E) (red bars, mean; $n > 2,000$ cells per condition). See also Appendix Fig S6L.
G   U2OS/GFP-FAM111A cell lines that were treated or not with DOX were stained with RFC1 antibody, pre-extracted and fixed, and stained with DAPI. RFC1 and DAPI signal intensities were analyzed by QIBC.
H   As in (G), except that cells were stained with RPB1 antibody.
I   Immunoblot analysis of U2OS/GFP-FAM111A cell lines treated or not with DOX in the absence or presence of Z-VAD-FMK.
J   As in (I), using U2OS cell lines conditionally expressing ectopic untagged FAM111A alleles.
K   Purified recombinant FLAG-FAM111A proteins were incubated at indicated temperatures for 4 h, and FAM111A auto-proteolytic activity was analyzed by immunoblotting.
L   GFP IPs from U2OS cell lines expressing GFP-FAM111A WT or D528G mutant were analyzed by mass spectrometry. Volcano plot shows enrichment of individual proteins (WT/D528G ratio) plotted against the $P$ value. Dashed lines indicate the significance thresholds (FDR $< 0.05$; $s_0 = 1$).

Data information: Data are representative of at least three (B–K) and two (L) independent experiments with similar outcomes.

phenotypes by amplifying its catalytic activity. Supporting this notion, FAM111A cleavage fragments that required its intrinsic protease activity but were insensitive to Caspase inhibition by Z-VAD-FMK were observed for disease-associated mutants but not the WT protein (Fig 4I and J). Importantly, using purified recombinant proteins we observed elevated auto-proteolytic cleavage of a FAM111A disease mutant *in vitro* (Figs 4K, and EV1C and D), providing direct evidence that patient-associated FAM111A mutations amplify its protease activity. In contrast, introducing a disease mutation had negligible impact on the subcellular localization of FAM111A and its association with interacting proteins (Figs 4L and EV4G). While dominant mutations in FAM111A underlie Kenny–Caffey syndrome, a variant form of this disorder can also be caused by recessive mutations in the tubulin-specific chaperone TBCE (Parvari *et al*, 2002). Similar to elevated FAM111A protease activity, we found that loss of TBCE expression led to impaired cell survival, and FAM111A patient mutants perturbed microtubule organization in a protease-dependent manner (Appendix Fig S3A–F), reminiscent of the impact of disease-associated *TBCE* mutations (Parvari *et al*, 2002), further implicating deregulated FAM111A protease activity in Kenny–Caffey syndrome etiology. Taken together, the data suggest that patient-associated point mutations in human FAM111A exert

their adverse impact on cell and organismal fitness by hyperactivating its intrinsic protease activity.

## Disease-associated FAM111B mutants have amplified proteolytic activity and phenocopy FAM111A-induced cellular phenotypes

Heterozygous missense mutations in human *FAM111B* (Fig 5A) are causative of a Rothmund–Thomson-like syndrome, whose molecular basis has not been explored (Khumalo *et al*, 2006; Mercier *et al*, 2013, 2015). As for FAM111A, all known patient-associated FAM111B mutations are found near the protease domain boundaries and are predicted to be surface-exposed based on our *in silico* modeling analysis (Figs 1E and 5A), suggesting possible commonalities in the mechanisms underlying their disease-promoting potential. Interestingly, our FAM111A interactome analysis revealed FAM111B as a candidate FAM111A-binding protein (Fig 2G; Dataset EV1). Indeed, an association between FAM111A and FAM111B in cells could be readily detected by reciprocal co-immunoprecipitation analysis, and *in vitro* binding experiments with purified proteins demonstrated that this interaction is direct (Fig 5B–D). The cellular function of FAM111B is unknown, but unlike FAM111A it lacks a recognizable PIP box and shows no detectable enrichment at DNA

**Figure 5. Disease-associated FAM111B mutants have amplified proteolytic activity and phenocopy FAM111A-induced cellular responses.**                                            ▶

A   Domain organization of human FAM111B and overview of disease-associated heterozygous missense mutations.
B   U2OS/GFP-FAM111A WT cells treated or not with DOX were subjected to GFP IP followed by immunoblotting with indicated antibodies.
C   As in (B), using U2OS/GFP-FAM111B WT cells.
D   Purified recombinant FLAG-FAM111A and FLAG-FAM111B proteins were incubated separately or mixed, subjected to IP with FAM111A antibody, and immunoblotted with FLAG antibody.
E   Immunoblot analysis of U2OS/GFP-FAM111B cell lines treated or not with DOX in the absence or presence of Z-VAD-FMK.
F   Cells in (E) were pulse-labeled with EdU, stained with DAPI, and analyzed for DAPI and EdU signal intensity using QIBC.
G   Quantification of data in (F) for S phase (EdU-positive) cells (red bars, mean (A.U., arbitrary units); $n > 2,000$ cells per condition). See also Appendix Fig S6N.
H   As in (F), except that cells were pulse-labeled with EU.
I   Quantification of EU incorporation in cells in (H) (red bars, mean; $n > 2,000$ cells per condition).
J   U2OS/GFP-FAM111B cell lines (Fig 5E) were fixed at the indicated times after DOX treatment and stained with crystal violet.
K   Cells in (E) were fixed, immunostained with γ-H2AX antibody, and subjected to QIBC analysis of γ-H2AX signal intensity (red bars, mean; $n > 2,000$ cells per condition). See also Appendix Fig S6O.
L   Immunoblot analysis of purified recombinant FLAG-FAM111B proteins and their auto-cleavage products.

Data information: Data (B–L) are representative of at least three independent experiments with similar outcomes.

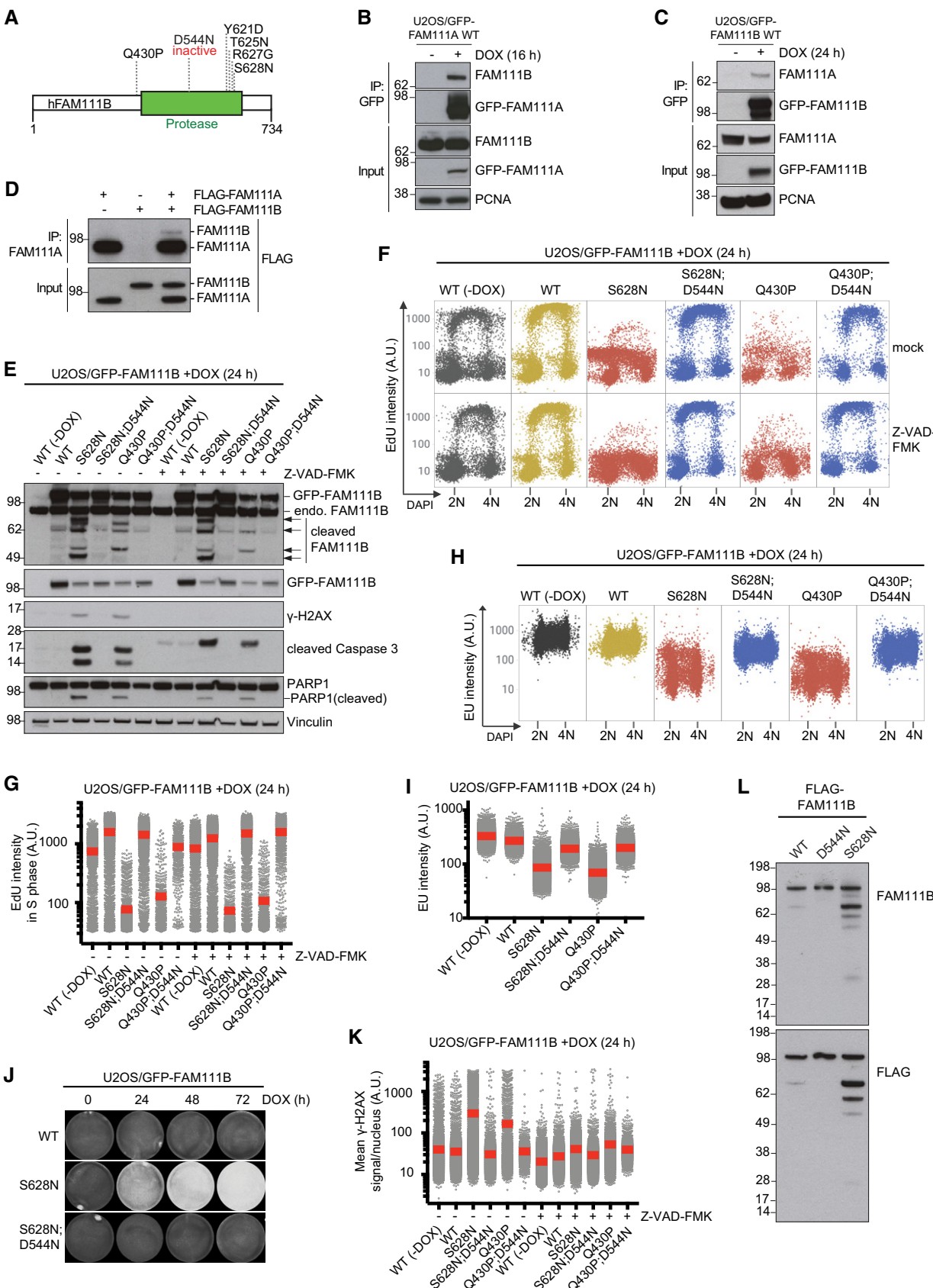

**Figure 5.**

replication sites (Fig EV5A; Wessel *et al*, 2019). Similar to FAM111A, we found that FAM111B was dispensable for normal DNA replication, PCNA chromatin loading, and cell proliferation (Appendix Fig S2A–H). Remarkably, however, using cell lines expressing GFP-tagged disease-associated FAM111B alleles at endogenous levels (Fig 5E and Appendix Fig S4A) we found that despite *FAM111A* and *FAM111B* mutations are associated with different syndromes, FAM111B patient mutants phenocopied the ability of elevated FAM111A protease activity to potently trigger replication and transcription shutdown, disruption of microtubule network integrity, and cell death via Caspase-dependent apoptosis (Fig 5E–K; Appendix Fig S4B–D). This suggests that FAM111A and FAM111B have at least partially overlapping cellular functions, consistent with these proteases forming a joint complex. In line with this, FAM111B also interacted with RFC subunits and RPB1, and the expression of a FAM111B disease mutant promoted dissociation of RFC1, PCNA, and RPB1 from chromatin (Fig EV5B–H). Unlike FAM111A, however, these phenotypes were only induced by the expression of patient-associated FAM111B alleles but not the WT protein in our cell lines (Figs 5E–K, and EV5B and E–H; Appendix Fig S4A–D). Consistently, FAM111B disease mutants showed prominent Caspase-independent formation of faster-migrating cleavage fragments that were not observed for WT FAM111B (Fig 5E; Appendix Fig S4C). Importantly, an inactivating D544N substitution in the catalytic protease domain of FAM111B disease mutants abolished their impact on DNA synthesis, transcription, microtubule organization, and apoptosis induction, and prevented the appearance of FAM111B proteolytic cleavage products (Figs 5E–K and EV5E–H; Appendix Fig S4D). Moreover, a disease-associated FAM111B S628N mutant displayed markedly elevated auto-proteolytic activity *in vitro* (Fig 5L, and EV1E). Thus, similar to FAM111A, patient mutations in FAM111B undermine cellular fitness by eliminating inhibitory constraints on its intrinsic protease activity, imposing a block to DNA and RNA synthesis, and promoting apoptotic cell death. However, despite FAM111A and FAM111B form a complex, their adverse impact on DNA-associated processes was not interdependent, as elevated FAM111A protease activity triggered these phenotypes in FAM111B-depleted cells and *vice versa* (Appendix Fig S5A–F).

Our findings suggest a common gain-of-function mechanism for how dominant mutations in FAM111A and FAM111B lead to multisystem disorders, driven by illegitimate amplification of their intrinsic proteolytic activities, which are detrimental to cellular fitness. Accordingly, identifying the targets of FAM111 protease activity will be an important task for the future. Although both RFC1 and RPB1 emerge as attractive candidate FAM111 substrates based on our findings, we have so far been unable to conclusively establish such relationships. While the biological functions of FAM111 proteases are not well-understood, our finding that FAM111 proteolytic activity is dispensable for normal cell growth but powerfully suppresses DNA replication, transcription, and cell viability once elevated is well-aligned with the putative role of FAM111A as a host restriction factor suppressing simian virus 40 (SV40) and poxvirus gene expression and DNA replication (Fine *et al*, 2012; Panda *et al*, 2017; Tarnita *et al*, 2019). When properly controlled, these features could provide an important contribution toward a general line of defense against invading viruses and their propagation. Given the strikingly similar cellular consequences of amplified FAM111A and FAM111B proteolytic activity, their homologous protease domains, and direct interaction, it seems likely that FAM111B may also act as a host restriction factor via a FAM111A-like mechanism of action.

In keeping with the notion that endogenous levels of patient-associated FAM111A and FAM111B mutants are sufficient to powerfully antagonize cellular fitness, the markedly different tissue expression profiles of *FAM111A* and *FAM111B* mRNAs (Appendix Fig S5G) offer a plausible rationale for why mutations in these proteases have distinct clinical manifestations despite their comparable cellular impacts. It is also possible that the cellular functions and substrates of FAM111A and FAM111B do not fully overlap. Our collective results suggest a potential of specific inhibitors of FAM111A or FAM111B catalytic activity as a rational treatment strategy for patients with fibrosing poikiloderma, gracile bone dysplasia, or Kenny–Caffey syndrome caused by mutations in these proteases. Moreover, they provide a foundation for further dissection of the clinical heterogeneity and precise molecular underpinnings of human syndromes caused by mutations in FAM111A and FAM111B, which may help to shed more light on the physiological functions of these proteases.

## Materials and Methods

### Plasmids and siRNAs

Full-length human *FAM111A* and *FAM111B* cDNAs were inserted into the destination vectors pcDNA4/TO/EGFP or pcDNA4/TO using Gateway LR Clonase (Invitrogen) to allow for doxycycline-inducible expression of the transgenes. Constructs encoding FLAG-tagged RFC1 were generated by inserting full-length human *RFC1* cDNA into pcDNA3.1+/FLAG and pFLAG-CMV2 vectors (containing C- and N-terminal FLAG tags, respectively). For expression in yeast, full-length human *FAM111A* and *FAM111B* cDNAs were cloned into the pYES-FLAG-SNAP-TopII vector. RFC1 deletion fragments were amplified by PCR and inserted into pcDNA3.1+/FLAG vector. Full-length human *RFC2, RFC3, RFC4,* and *RFC5* cDNAs were inserted into the destination vector pcDNA4/TO/HA-FLAG. Point mutations in FAM111A and FAM111B were introduced using the QuikChange Site-Directed Mutagenesis Kit (Agilent Technologies) according to the manufacturer's protocol. The FAM111A *PIP mutant (Alabert *et al*, 2014) was generated by introducing Y24A and F25A substitutions into full-length FAM111A. Plasmids for generation of cell lines with targeted knockout of FAM111A and/or FAM111B (*ΔFAM111A, ΔFAM111B,* and *ΔFAM111A + ΔFAM111B*) using CRISPR/Cas9 were constructed as described (Cong *et al*, 2013), using the pX459 plasmid (Addgene #62988) for Cas9 expression and sgRNA delivery. Briefly, sgRNA sequences were ordered as complementary primers, mixed in a 1:1 ratio, and annealed. Subsequently, pX459 was digested with BbsI and the sgRNA introduced using a normal ligation reaction according to the manufacturer's instructions (New England Biolabs). The following sequences were used: *FAM111A* sgRNA #2 (forward): 5′-CACCGAAGAGCCACAACTAATACCC-3′; *FAM111A* sgRNA #2 (reverse): 5′-AAACGGGTATTAGTTGTGGCT CTTC-3′; *FAM111B* sgRNA #2 (forward): 5′-CACCGTAAAC

TCACAAGTTAGACGG-3′; and *FAM111B* sgRNA #2 (reverse): 5′-AAACCCGTCTAACTTGTGAGTTTAC-3′.

Plasmid DNA and siRNA transfections were performed using FuGENE 6 Transfection Reagent (Promega) and Lipofectamine RNAiMAX (Invitrogen), respectively, according to the manufacturers' protocols. All siRNAs were used at a final concentration of 50 nM. The following siRNA oligonucleotides were used: non-targeting control (CTRL): 5′-GGGAUACCUAGACGUUCUA-3′; FAM111A (#3): 5′-GGAGAAUGAUGAUUGGAAA-3′; FAM111A (#5): 5′-GGGAAGAAUAACAAGAUUA-3′; FAM111B (#1): 5′-CCUGUU GAUCAUUGUCUAU-3′; p53 (#1): 5′-CUACUUCCUGAAAACAACG-3′; p53 (#2): 5′-GAAAUUUGCGUGUGGAGUA-3′; p53 (#3): 5′-GACUC-CAGUGGUAAUCUAC-3′; RFC1: 5′-GGUAUGAGCAGUAGGCUUA-3′; TBCE (#1): 5′-CAGACUUUCUUACUGCAAU-3′; and TBCE (#2): 5′-CCUUGAGUCUAACAACAUU-3′.

## Cell culture

Human U2OS and HCT116 cells were obtained from ATCC. All cell lines used in this study were cultured in DMEM containing 10% FBS and were regularly tested negative for mycoplasma infection. To generate U2OS or HCT116 cell lines inducibly expressing GFP-tagged or untagged FAM111A as well as GFP-FAM111B WT and mutant alleles, cells were co-transfected with pcDNA4/TO/GFP or pcDNA4/TO expression constructs and pcDNA6/TR (Invitrogen). Positive clones were selected by incubation in medium containing 5 μg/ml blasticidin S (Invitrogen) and 400 μg/ml Zeocin (Invitrogen) for 14 days. To generate cell lines with targeted knockout of FAM111A and/or FAM111B (*ΔFAM111A, ΔFAM111B*, and *ΔFAM111A + ΔFAM111B*), parental U2OS cells were transfected with pX459-sgFAM111A#2 (*ΔFAM111A*), pX459-sgFAM111B#2 (*ΔFAM111B*), or a 1:1 mix of pX459-sgFAM111A#2 and pX459-sgFAM111B#2 (*ΔFAM111A + ΔFAM111B*) and selected briefly with puromycin during clonal selection. Clones were screened for FAM111A and FAM111B expression by immunoblotting. Unless otherwise indicated, the following drug concentrations were used: nocodazole (10 μM, Sigma-Aldrich), Z-VAD-FMK (50 μM, ab120487, Abcam), doxycycline (1 μg/ml, Sigma-Aldrich), and AEBSF (5 mM, Sigma-Aldrich).

## Immunochemical methods

For immunoblotting and immunoprecipitation, which were performed as previously described (Poulsen *et al*, 2012), cells were lysed in EBC buffer (50 mM Tris, pH 7.5; 150 mM NaCl; 1 mM EDTA; 0.5% NP40; 1 mM DTT) supplemented with phosphatase inhibitors and protease inhibitor cocktail (Roche). Lysates were then incubated for 10 min on ice and sonicated. For immunoprecipitations, cleared lysates were incubated with FLAG agarose (Sigma-Aldrich), GFP-Trap Agarose (Chromotek), or anti-goat RFC1 antibody (2 μg/sample) coupled to Protein G agarose beads (Thermo Fisher Scientific) for 2 h on an end-over-end rotator at 4°C, washed in EBC buffer, and treated with Benzonase to minimize chromatin-mediated interactions. Proteins were resolved by SDS–PAGE and analyzed by immunoblotting. For *in vitro* binding assays, recombinant proteins were diluted in EBC buffer and incubated with FAM111A antibody (2 μg/sample) coupled to Protein A Agarose beads (Thermo Fisher Scientific) at 4°C for 2 h on a rotary wheel,

followed by washes in EBC buffer. For *in vitro* auto-cleavage assays, recombinant proteins were incubated in EBC buffer.

## Immunofluorescence and high-content imaging analysis

Where indicated, cells were pre-extracted in PBS containing 0.2% Triton X-100 for 2 min on ice, before fixation with 4% formaldehyde for 15 min. If cells were not pre-extracted, they were subjected to a permeabilization step with PBS containing 0.2% Triton X-100 for 5 min and incubated with primary antibodies diluted in 1% BSA-PBS for 1 h at room temperature. Following staining with secondary antibodies (Alexa Fluor; Life Technologies) and 4′,6-diamidino-2-phenylindole dihydrochloride (DAPI, 0.5 μg/ml, DNA staining) diluted in 1% BSA-PBS for 1 h at room temperature, cells were mounted onto glass slides using ProLong Gold Antifade (Invitrogen).

For EdU stainings, cells were treated with EdU (10 μM) for 30 min before fixation and then stained using the Click-iT Plus EdU Alexa Fluor 647 Imaging Kit (Invitrogen) according to the manufacturer's instructions before incubation with primary antibodies. For EU stainings, cells were treated with EU (1 mM) for 60 min before fixation and then stained using the Click-iT RNA Alexa Fluor 594 Imaging Kit (Invitrogen) according to the manufacturer's instructions.

Images were acquired with a Leica AF6000 wide-field microscope (Leica Microsystems) equipped with HC Plan-Apochromat 63×/1.4 oil immersion objective, using standard settings. Image acquisition and analysis were carried out with Leica Application Suite X software (version 3.3.3.16958; Leica Microsystems). Raw images were exported as TIFF files, and if adjustments in image contrast and brightness were applied, identical settings were used on all images of a given experiment. Quantitative image-based cytometry (QIBC) was performed as described (Toledo *et al*, 2013). Briefly, cells were fixed, permeabilized, and stained as described above. Images were acquired with a scanR inverted high-content screening microscope (Olympus) equipped with wide-field optics, UPLSAPO dry objectives (×20, 0.75-NA), fast excitation and emission filter-wheel devices for DAPI, FITC (fluorescein isothiocyanate), Cy3 and Cy5 wavelengths, an MT20 illumination system, and a digital monochrome Hamamatsu ORCA-R2 CCD camera. Automated and unbiased image analysis was carried out with the ScanR analysis software (version 2.7.1). Data were exported and processed using Spotfire software (version 10.5.0; Tibco).

## Antibodies

Antibodies used for immunoblotting included acetylated α-tubulin (acetyl K40, ab24610, Abcam (1:5,000 dilution)), actin (MAB1501, Millipore (1:20,000)), Cleaved Caspase-3 (Asp175, 9661, Cell Signaling (1:1,000)), CTCF (A300-543A, Bethyl (1:500)), FAM111A (ab184572, Abcam (1:1,000)), FAM111B (HPA038637, Sigma (1:2,000)), FLAG (A00187, GenScript (1:1,000)), GFP (11814460001, Roche (1:500)) or sc-8334, Santa Cruz (1:1,000)), GTFC1 (A301-292A, Bethyl (1:1,000)), histone H2AX (2595, Cell Signaling (1:1,000)), IFI-16 (sc-8023, Santa Cruz (1:500)), MCM2 (610701, Clone 46/BM28, BD Transduction Lab (1:1,000)), P53 (sc-126, Santa Cruz (1:1,000)), PARP-1 (sc8007, Santa Cruz (1:500)), PCNA (sc-56, Santa Cruz (1:1,000)), RFC1 (ab3566, Abcam (1:1,000)), RFC3

(ab154899, Abcam (1:1,000)), RFC5 (A300-146A, Bethyl (1:1,000)), TBCE (A305-485A, Bethyl (1:1,000)), tubulin alpha (T9026, Sigma-Aldrich (1:5,000)), and vinculin (V9131, Sigma (1:10,000)). Antibodies used for immunoprecipitation (IP) included FAM111A (ab184572, Abcam, 2 µg/IP) and RFC1 (ab3566, Abcam, 2 µg/IP). Antibodies used for immunofluorescence included γH2AX (05-636 (Clone JBW301), Millipore (1:500)), FAM111A (ab184572, Abcam (1:300)), MCM2 (610701, Clone 46/BM28, BD Transduction Lab (1:500)), PCNA (#2037, Triolab Immunoconcepts (1:500)), RFC1 (sc-271656, Santa Cruz (1:300) or ab3566, Abcam (1:1,000)), RPA2 (NA19L (Clone Ab-3), Roche (1:1,000)), and α-tubulin (T9026, Sigma-Aldrich (1:5,000)).

## DNA fiber assays

Exponentially growing U2OS cells ($1 \times 10^6$) were labeled with consecutive pulses of CldU (25 µM) and IdU (250 µM) for 25 min. Cells were then trypsinized and resuspended in PBS. Cell suspension (2 µl) was spotted onto Superfrost glass slides and lysed in buffer containing 200 mM Tris–HCl, pH 7.4; 50 mM EDTA; and 0.5% SDS for 2 min. Slides were tilted at an angle to allow the DNA to run slowly down the slide and air-dried before fixation in 3:1 methanol:acetic acid. DNA fiber spreads were denatured with 2.5 M HCl for 90 min before blocking in 2% BSA-PBS with 0.1% Tween for 30 min. Slides were then incubated with rat anti-BrdU (Abcam, ab6326; 1:100 dilution) for 1 h to detect CldU. Slides were washed in PBS-Tween and PBS before antibody cross-linking in 4% formaldehyde for 10 min. Slides were then incubated with Alexa Fluor 594 goat anti-rat antibody (Thermo Fisher; 1:100) for 1 h. Following similar washes, slides were incubated with mouse anti-BrdU (BD Bioscience, #347580; 1:500) overnight at 4°C to detect IdU. Slides were washed and incubated with Alexa Fluor 488 goat anti-mouse antibody (Thermo Fisher; 1:100) for 1 h. After washing, the slides were air-dried and mounted with 50 µl VECTASHIELD mounting medium (Vector Laboratories). Track lengths were measured using ImageJ software.

## Proliferation and cell survival assays

Cell lines were seeded in 96-well plates in triplicates, and cell proliferation was determined for 4 days by incubation with 10 µg/ml of resazurin (Sigma) for 2 h at 37°C. Fluorescence was measured at 590 nm using a plate reader (FLUOstar® Omega, BMG Labtech). The obtained values were normalized to the values of the first day. For survival assays, cells were seeded in 6-cm dishes and treated with doxycycline to induce the expression of ectopic FAM111A or FAM111B alleles for the indicated times. After 72 h, plates were washed once in PBS, left to dry, and stained with cell staining solution (0.5% *w/v* crystal violet, 25% *v/v* methanol). Finally, plates were washed three times in deionized water. Images were acquired using the GelCount™ (Oxford Optronix) colony counter.

## Affinity purification and mass spectrometry (AP-MS)

Partial on-bead digestion was used for peptide elution from GFP-Trap Agarose (Chromotek). Briefly, 100 µl of elution buffer (2 M urea; 2 mM DTT; 20 µg/ml trypsin; and 50 mM Tris, pH 7.5) was added and incubated at 37°C for 30 min. Samples were alkylated

with 25 mM CAA and digested overnight at room temperature before addition of 1% trifluoroacetic acid (TFA) to stop digestion. Peptides were desalted and purified with styrene–divinylbenzene reversed-phase sulfonate (SDB-RPS) StageTips. Briefly, two layers of SDB-RPS were prepared with 100 µl wash buffer (0.2% TFA in $H_2O$). Peptides were loaded on top and centrifuged for 5 min at 500 *g*, and washed with 150 µl wash buffer. Finally, peptides were eluted with 50 µl elution buffer (80% ACN and 1% ammonia) and vacuum-dried. Dried peptides were dissolved in 2% acetonitrile (ACN) and 0.1% TFA in water and stored at −20°C.

## Liquid chromatography–mass spectrometry (LC-MS) analysis

Nanoflow LC-MS analysis of tryptic peptides was performed using a quadrupole Orbitrap mass spectrometer [Q Exactive HF-X, Thermo Fisher Scientific (Kelstrup *et al*, 2018)] connected to an EASY-nLC 1200 ultra-high-pressure system (Thermo Fisher Scientific). Approximately 0.5 µg of peptides was loaded on a 50-cm HPLC column (75 µm inner diameter, New Objective; in-house packed using ReproSil-Pur C18-AQ 1.9 lm silica beads; Dr Maisch GmbH, Germany). Peptides were separated using a linear gradient from 2 to 20% B in 55 min and stepped up to 40% in 40 min followed by a 5 min wash at 98% B at 350 nl/min, where solvent A was 0.1% formic acid in water and solvent B was 80% acetonitrile and 0.1% formic acid in water for a total duration of 100 min. The mass spectrometer was operated in "top-15" data-dependent mode, collecting MS spectra in the Orbitrap mass analyzer (60,000 resolution, 300–1,650 *m/z* range) with an automatic gain control (AGC) target of $3 \times 10^6$ and a maximum ion injection time of 25 ms. The most intense ions from the full scan were isolated with an isolation width of 1.4 *m/z*. Following higher-energy collisional dissociation (HCD) with a normalized collision energy (NCE) of 27, MS/MS spectra were collected in the Orbitrap (15,000 resolution) with an AGC target of $1 \times 10^5$ and a maximum ion injection time of 28 ms. Precursor dynamic exclusion was enabled with a duration of 30 s.

## Bioinformatic analyses

Raw MS files were processed using the MaxQuant software (version 1.6.5.0) (Cox & Mann, 2008). The integrated Andromeda search engine (Cox *et al*, 2011) was used for peptide and protein identification at an FDR of < 1% and $s_0$ value of 1 (Tusher *et al*, 2001). Missing values were imputed based on a normal distribution (width = 0.15; downshift = 1.8). The human UniProtKB database (January 2019) was used as forward database and the automatically generated reverse database for the decoy search. A minimum number of seven amino acids was used for the peptide identification. Proteins that could not be discriminated by unique peptides were pooled in the same protein group (Cox & Mann, 2008). Label-free protein quantification was done using the MaxLFQ algorithm (Cox *et al*, 2014). Protein ratios were calculated based on median peptide ratios, and only common peptides were used for pairwise ratio calculations. The "match-between-runs" feature of MaxQuant was enabled to transfer peptide identifications across runs based on high mass accuracy and normalized retention times. All statistical and bioinformatic analyses were performed using Perseus (Tyanova *et al*, 2016) or the R framework (https://www.r-project.org/). The mass spectrometry proteomic data have been deposited to the

ProteomeXchange Consortium via the PRIDE (Perez-Riverol *et al*, 2019) partner repository with the dataset identifier PXD017978.

### Purification of recombinant FAM111 proteins

Identical conditions were used for purification of all FLAG-tagged recombinant FAM111A or FAM111B proteins. Briefly, JEL-1 yeast strain transformed with FLAG-FAM111A or FLAG-FAM111B expression plasmid was grown in SC-URA medium containing 2% glucose for 36 h at 30°C, and subsequently in SC-URA medium containing 2% raffinose for 24 h at 30°C. The culture was then grown in YEP medium containing 2% raffinose and grown until OD 0.8. FLAG-FAM111 protein expression was induced by adding galactose (2%) for 24 h at 20°C. Cells were collected and resuspended in lysis buffer containing 50 mM Tris, pH 7.5; 500 mM NaCl; 10% glycerol; 1 mM EDTA; and 0.1 mM DTT. Glass beads were added to the resuspended cells, which were lysed by vortexing and cleared by centrifugation at 25,000 *g* at 4°C for 30 min. The cleared lysate was incubated with FLAG M2 resin (Sigma-Aldrich) and incubated at 4°C for 2 h. After extensive washing, FLAG-FAM111A or FLAG-FAM111B was eluted in lysis buffer supplemented with FLAG peptide (0.5 mg/ml). The elute fractions were run on a 4–12% NuPAGE Bis-Tris protein gel (Invitrogen) and stained with Instant Blue Coomassie Protein Stain (Expedeon). Fractions were concentrated on Microcon-30 kDa Centrifugal Filters (Millipore), snap-frozen in liquid nitrogen, and stored at −80°C.

### Homology modeling of FAM111A and FAM111B protease domains

Domain identification in human FAM111A and FAM111B with Conserved Domains Database (CDD) (Lu *et al*, 2020; https://www.ncbi.nlm.nih.gov/Structure/cdd/wrpsb.cgi) identified residues 371–555 for FAM111A and residues 471–664 for FAM111B to contain conserved trypsin-like serine protease domains. Full-length FAM111A and FAM111B sequences were used as input in HHpred for the identification of experimental structures of homologous proteins (https://toolkit.tuebingen.mpg.de/tools/hhpred). Among obtained hits, the crystal structure of *E. coli* DegS protease (2R3U) at 2.6 Å resolution (Hasselblatt *et al*, 2007) displayed the highest degree of homology and was used as a template for generating FAM111A and FAM111B protease domain models using MODELLER (Zimmermann *et al*, 2018). The quality of the models was assessed based on the Ramachandran plot using Rampage (http://mordred.bioc.cam.ac.uk/~rapper/rampage.php), which assigned 85.7% of residues to favored regions (2.9% outliers) for FAM111A and 88.1% of residues to favored regions (4.5% outliers) for FAM111B. Overall model quality for both FAM111A (*Z*-score: −4.05) and FAM111B (Z-score: −3.82) was assessed using ProSA-web Protein Structure Analysis (https://prosa.services.came.sbg.ac.at/prosa.php). PyMol (v2.0) was used for visualization of the structures.

### Flow cytometry

Cells were fixed in 70% ethanol, and DNA was stained with propidium iodide (0.1 mg/ml) containing RNase (20 μg/ml) for 30 min at 37°C. Flow cytometry analysis was performed on a FACSCalibur (BD Biosciences) using CellQuest Pro software

(version 6.0; Becton Dickinson). The data were analyzed using FlowJo software (version 10.6).

### *FAM111* transcript profiling

Tissue RNA-seq data (RNA HPA tissue gene data) were downloaded from The Human Protein Atlas (Uhlen *et al*, 2015) (version 19.3) and Ensembl (version 92.38).

## Data availability

Mass spectrometry proteomic RAW data are available at the ProteomeXchange Consortium database via the Proteomics Identifications (PRIDE) partner repository (http://www.ebi.ac.uk/pride), under the dataset ID PXD017978. All other data supporting the findings of this study are available within the article and supplementary information.

**Expanded View** for this article is available online.

### Acknowledgements

We thank Blanca Lopez Mendez, Kata Sarlós, Veronika Baráth, and the Novo Nordisk Foundation Center for Protein Research imaging platform for providing reagents and technical assistance. This work was supported by Novo Nordisk Foundation (grants no. NNF14CC0001 and NNF18OC0030752), the Lundbeck Foundation (grant no. R303-2018-3212), European Research Council (ERC, grant agreement no. 616236 (DDRegulation)), Danish Cancer Society (grant no. R231-A13972), and Danish National Research Foundation (grant no. DNRF115).

### Author contributions

SH and NM conceived the project; SH carried out the majority of the experiments; SP, AM, PH, FC, and MG performed experiments; SH, SP, AM, PH, FC, NMIT, and NM designed experiments and analyzed the data; MM, NMIT and NM supervised the project; SH and NM acquired funding; NM wrote the manuscript with inputs from all authors.

### Conflict of interest

The authors declare that they have no conflict of interest.

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
