## [Review Process File · EMBO Reports]

FAM111 protease activity undermines cellular fitness and is amplified by gain-of-function mutations in human disease

Saskia Hoffmann, Satyakrishna Pentakota, Andreas Mund, Peter Haahr, Fabian Coscia, Marta Gallo, Matthias Mann, Nicholas Taylor, and Niels Mailand

DOI: [10.15252/embr.202050662](https://doi.org/10.15252/embr.202050662)

Corresponding author(s): Niels Mailand (niels.mailand@cpr.ku.dk)

Review Timeline:

Submission Date:	16th Apr 20
Editorial Decision:	15th May 20
Revision Received:	16th Jun 20
Editorial Decision:	6th Jul 20
Revision Received:	13th Jul 20
Accepted:	20th Jul 20

Transaction Report:

Dear Niels,

Thank you for the submission of your research manuscript to our journal. We have now received the full set of referee reports that is copied below.

As you will see, all three referees acknowledge that the findings are interesting and overall well documented but ask for the addition of a set of control experiments, statistical evaluation and textual changes, which should be provided.

Given these constructive comments, we would like to invite you to revise your manuscript with the understanding that the referee concerns (as detailed above and in their reports) must be fully addressed and their suggestions taken on board. Please address all referee concerns in a complete point-by-point response. Acceptance of the manuscript will depend on a positive outcome of a second round of review. It is EMBO reports policy to allow a single round of revision only and acceptance or rejection of the manuscript will therefore depend on the completeness of your responses included in the next, final version of the manuscript.

Revised manuscripts should be submitted within three months of a request for revision; they will otherwise be treated as new submissions. Please contact us if a 3-months time frame is not sufficient for the revisions so that we can discuss the revisions further.

We invite you to submit your manuscript within three months of a request for revision. This would be August 15th in your case. Yet, given the current COVID-19 related lockdowns of laboratories, we have extended the revision time for all research manuscripts under our scooping protection to allow for the extra time required to address essential experimental issues. Please contact us to discuss the time needed and the revisions further.

Before I list the general instructions for submitting the revised manuscript, I will list a few items specific to your manuscript that should be addressed.

1) Supplementary material: Please change the nomenclature to Appendix Figure Sx and call the pdf file "Appendix". The Appendix needs a title page with a table of content and page numbers. The legend for table S1 should be removed from the pdf file. Table S1 should be submitted as Dataset EV1 in the form of an excel file containing the legend, e.g., in the first tab.

2) Data availability section: please add a link that resolves to the database and provide reviewer access. This is a formal requirement for our quality control that happens when the revision is in.

3) Whenever you display quantification, please keep mind to specify in the figure legend: the nature of the bars and error bars, the statistical test used, and the number of replicates the quantification is based on. Please also specify if these are biological or technical replicates. In case $n < 3$ please display the data in the form of scatter blots.

General instructions: When submitting your revised manuscript, we will require:

2) individual production quality figure files as .eps, .tif, .jpg (one file per figure).

Please download our Figure Preparation Guidelines (figure preparation pdf) from our Author Guidelines pages

<https://www.embopress.org/page/journal/14693178/authorguide> for more info on how to prepare your figures.

4) a complete author checklist, which you can download from our author guidelines (). Please insert information in the checklist that is also reflected in the manuscript. The completed author checklist will also be part of the RPF.

5) Please note that all corresponding authors are required to supply an ORCID ID for their name upon submission of a revised manuscript (). Please find instructions on how to link your ORCID ID to your account in our manuscript tracking system in our Author guidelines

()

6) We replaced Supplementary Information with Expanded View (EV) Figures and Tables that are collapsible/expandable online. A maximum of 5 EV Figures can be typeset. EV Figures should be cited as 'Figure EV1, Figure EV2" etc... in the text and their respective legends should be included in the main text after the legends of regular figures.

7) Data availability section:

The accession numbers and database should be listed in a formal "Data Availability " section (placed after Materials & Method) that follows the model below (see also <<https://www.embopress.org/page/journal/14693178/authorguide#dataavailability>>). Please note that the Data Availability Section is restricted to new primary data that are part of this study.

Data availability

- RNA-Seq data: Gene Expression Omnibus GSE46843
(<https://www.ncbi.nlm.nih.gov/geo/query/acc.cgi?acc=GSE46843>)

- [data type]: [name of the resource] [accession number/identifier/doi] ([URL or identifiers.org/DATABASE:ACCESSION])

8) We would also encourage you to include the source data for figure panels that show essential data. Numerical data should be provided as individual .xls or .csv files (including a tab describing the data). For blots or microscopy, uncropped images should be submitted (using a zip archive if multiple images need to be supplied for one panel). Additional information on source data and instruction on how to label the files are available .

10) Regarding data quantification:

- Please ensure to specify the name of the statistical test used to generate error bars and P values, the number (n) of independent experiments underlying each data point (not replicate measures of one sample), and the test used to calculate p-values in each figure legend. Discussion of statistical methodology can be reported in the materials and methods section, but figure legends should contain a basic description of n, P and the test applied.

IMPORTANT: Please note that error bars and statistical comparisons may only be applied to data obtained from at least three independent biological replicates. If the data rely on a smaller number of replicates, scatter blots showing individual data points are recommended.

- Graphs must include a description of the bars and the error bars (s.d., s.e.m.).

11) As part of the EMBO publication's Transparent Editorial Process, EMBO reports publishes online a Review Process File to accompany accepted manuscripts. This File will be published in conjunction with your paper and will include the referee reports, your point-by-point response and all pertinent correspondence relating to the manuscript.

I look forward to seeing a revised version of your manuscript when it is ready. Please let me know if you have questions or comments regarding the revision.

Kind regards,

Martina

Martina Rembold, PhD
Editor
EMBO reports

Referee #1:

EMBOR-2020-5066

This manuscript by Hoffmann and colleagues reports on the protein paralogs FAM111A and FAM111B. These two proteins are predicted to contain trypsin-like serine protease domains, have been implicated as viral restriction factors, and present as heterozygote dominant mutants in several poorly characterized diseases. In addition, several reports have implicated FAM111B in chromatin binding, PCNA binding, and DNA replication regulation although most reports have been incomplete. The current report represents a significant advance by reporting that FAM111A and FAM111B can bind to each other, overexpression causes decreased DNA replication and RNA transcription, and the disease associated mutants have increased protease activity with phenotypic behavior similar but stronger to the wild type alleles. Overall this report brings interesting new insights into these proteins that may help to understand the mutant activity. Certain controls would improve the quality of the report

Major

1. The authors make the observation that inducible expression of GFP-tagged FAM111A but not FAM111B induces a growth arrest (Figure 1I), decreased EdU incorporation (Fig. 2C), decreased RFC1 binding to chromatin (Fig. 2J), EU incorporation (Fig. 3F). However, in Figure 4, they report that inducible WT (low) does not induce similar effects. It would be very helpful to show in the same experiment that the levels of inducible wild type FAM111A are different in these two different cells and that the incorporation of EdU (and other markers) is affected by the high levels of FAM111A but not low. If it is not simply differences in levels could this be due to other clonal differences in the cell lines?

2. In Figure 5J, overexpression of a disease-associated mutant form of FAM111B induces growth arrest but not the same mutant allele containing a substitution mutation disabling the protease domain. It would be useful to show in the same experiment that wild type FAM111B does not induce a growth inhibitory phenotype as was shown in Fig 1I.

3. Several figures show a faster migrating product of FAM111A and FAM111B with prior incubation at 37°C (Fig. 1G, 1H, 4J, 4K). While inhibition of caspase 3 by Z-VAD-FMK does not reduce the appearance of the reported cleaved product, it would be useful to show that a specific serine protease inhibitor reduces the appearance of this band at the elevated temperature. While substitution in the proposed active site (D439N) reduces the signal, a specific protease inhibitor would support that the activity in the wild type and disease associated mutant forms have self-cleavage protease activity. It would also be helpful to indicate where the epitope for the antibody is located that can recognize the full-length and cleaved form.

Minor

1. The authors predict the structure of FAM111A and FAM111B by modeling it on the solved crystal structure of the bacterial protease DegS. However, it is not always clearly written in the text that the predicted location of various residues in FAM111A/B are predicted and not solved. For example, the various disease-associated mutant residues are predicted to be located outside of the core protease domain.

Referee #2:

This manuscript describes FAM111A and FAM111B as proteases that are hyperactivated by disease causing mutations. Their overexpression or hyperactivation yields defects in several biological processes including replication, transcription, and microtubule organization. The data convincingly demonstrates that the disease-causing mutations are gain of function and provide some mechanistic explanations for the biological consequences such as interaction with and displacement of the RFC protein complex from chromatin. Overall, the data is very high quality and the conclusions are appropriate. I offer the following comments to the authors for their consideration to improve their interesting and well-written manuscript.

1. Figure 1I: Was the level of FAM111A and FAM111B expression similar? How about the mutants? Documentation would be useful in this figure since the authors do compare effects. The authors may also want to comment in the text on why the protease catalytic mutants are often overexpressed at higher levels (presumable selection since they are less toxic to the cells).

2. FAM111A was reported to be enriched at replication forks in Wessel et al., Cell Reports 2019 (<https://doi.org/10.1016/j.celrep.2019.08.051>) and its knockdown was reported in that paper to cause hypersensitivity to ATR inhibitors suggesting a function in replication. I recommend adding this paper to the references on page 5 where the Groth Nascent chromatin capture reference is utilized. Interestingly, FAM111B was not identified as a fork-associated protein. While negative data that is difficult to interpret, this is somewhat surprising given the authors documentation that 111A and 111B interact and overexpression of either causes similar problems during replication. The authors may want to comment on this in relation to their observation that only the disease causing 111B mutant proteins generate the same level of replication problems.

Referee #3:

Elucidating the functions of proteins involved in clinical syndromes is a vital part in understanding how to develop treatments. Hoffmann et al. report both the function of FAM111A and B, as well as the significance of patient mutations in these genes leading to a broad spectrum of disease manifestations. The authors first show that human FAM111A and FAM111B are active proteases in vitro and when FAM111A is overexpressed it substantially lowers cell viability through inhibition of replication and transcription, induction of DNA damage response and activation of apoptosis. They also show that the patient-associated mutations in both proteins result in overactive proteases and they phenocopy the overexpression of FAM111A. To identify a mechanism of the profound effect on replication, the authors show via an interactome analysis as well as various co-IP experiments, that FAM111A interacts with RFC1, and that increased FAM111A protease activity leads to hindered DNA replication by causing RFC1 and PCNA displacement from chromatin.

Overall, this is an excellent manuscript that should be published with minor modifications. The major issue, which the authors are aware of, is that they do not have an actual substrate list for the two proteases and ultimately this is what it would take to understand the plethora of phenotypes associated with the mutations. However, the manuscript has a beautiful set of experimental evidence about the function of the proteases that will be a starting point to fully understand the protease function and the diseases associated with their overactivation.

One issue that should be addressed before publication is the lack of statistical significance indication in any of the dot-plots. That information should be included even if the differences are obvious.

Other points:

-The authors identify RFC1, 2, 5 as enriched (Figure 2G). If they do not want to reveal the whole list, can they show some GO analysis of the enriched proteins?

-The authors mention that the literature shows FAM111A is important for loading PCNA. Then contradict the literature by saying that with CRISPR and siRNA of FAM111A, there's no impact on DNA replication, without showing any data on how the KD/KO of FAM111A impacts levels of PCNA. PCNA loading and DNA replication are not interchangeable terms. They should show PCNA levels when FAM111A is KD/KO similarly to Fig 2C if they have it.

- "These findings suggest that FAM111A inhibits DNA replication by promoting RFC and PCNA displacement from chromatin via its intrinsic protease activity" seems like an overstatement. Ultimately, to make this conclusion, more work would be needed on replacing the missing RFC and PCNA to the chromatin to see if proper replication could be restored. Overexpressing these proteases likely affects many other proteins as well (besides just RFC and PCNA), ultimately leading to replication defect and increased DNA damage.

Point-by-point reply to the referees' comments

We would like to thank the referees for the constructive and insightful comments and suggestions they made on our manuscript. We were delighted to see that all three referees were enthusiastic about our study and were supportive in principle of publication in EMBO Reports. In the revised manuscript, we included the results of several new experiments performed on the basis of the reviewers' helpful suggestions, and we clarified a number of points in the text. Collectively, we believe the new additions and changes to the manuscript address all of the referees' concerns, as explained in the detailed point-by-point response to the referee reports (replicated in full) below.

Referee #1:

This manuscript by Hoffmann and colleagues reports on the protein paralogs FAM111A and FAM111B. These two proteins are predicted to contain trypsin-like serine protease domains, have been implicated as viral restriction factors, and present as heterozygote dominant mutants in several poorly characterized diseases. In addition, several reports have implicated FAM111B in chromatin binding, PCNA binding, and DNA replication regulation although most reports have been incomplete. The current report represents a significant advance by reporting that FAM111A and FAM111B can bind to each other, overexpression causes decreased DNA replication and RNA transcription, and the disease associated mutants have increased protease activity with phenotypic behavior similar but stronger to the wild type alleles. Overall this report brings interesting new insights into these proteins that may help to understand the mutant activity. Certain controls would improve the quality of the report

Major

1. The authors make the observation that inducible expression of GFP-tagged FAM111A but not FAM111B induces a growth arrest (Figure 1I), decreased EdU incorporation (Fig. 2C), decreased RFC1 binding to chromatin (Fig. 2J), EU incorporation (Fig. 3F). However, in Figure 4, they report that inducible WT (low) does not induce similar effects. It would be very helpful to show in the same experiment that the levels of inducible wild type FAM111A are different in these two different cells and that the incorporation of EdU (and other markers) is affected by the high levels of FAM111A but not low. If it is not simply differences in levels could this be due to other clonal differences in the cell lines?

Figure EV4A compares the levels of GFP-FAM111A expression in the WT and WT (low) cell lines side-by-side. As can be seen, the GFP-FAM111A expression level in WT (low) cells (1.7-fold that of endogenous FAM111A) is somewhat lower than in the WT cell line (2.5-fold higher expression than endogenous FAM111A). To more conclusively show that the differential impact of FAM111A induction on DNA replication in the WT and WT (low) cell lines is due to differences in the expression levels of ectopic FAM111A, we included new data comparing the impact of ectopic FAM111A expression on EdU incorporation in these cell lines at different time points. As can be seen from these experiments, the WT (low) cells show less pronounced inhibition of DNA replication at 16 h after induction than WT cells (new Figure EV4B). However, at 24 h after induction of the transgenes, DNA replication is fully suppressed in both the WT and WT (low) cell lines (new Figure EV4C). These data show that GFP-FAM111A WT induction in both cell lines inhibits DNA replication, yet the WT (low) cells do so with a slower kinetics due to their lower level of FAM111A expression. We now explain this more carefully in the text (page 8).

2. In Figure 5J, overexpression of a disease-associated mutant form of FAM111B induces growth arrest but not the same mutant allele containing a substitution mutation disabling the protease domain. It would be useful to show in the same experiment that wild type FAM111B does not induce a growth inhibitory phenotype as was shown in Fig 1I.

We agree that showing the impact of WT and mutant FAM111B alleles on overall cell growth in different figures (previously Figure 1I and Figure 5J) was not an ideal way to present these data. In the revised manuscript, we therefore display the results for all FAM111B cell lines (WT, S628N and S628N/D544N) in a single panel (Figure 5J).

3. Several figures show a faster migrating product of FAM111A and FAM111B with prior incubation at 37 °C (Fig. 1G, 1H4I 4J, 4K). While inhibition of caspase 3 by Z-VAD-FMK does not reduce the appearance of the reported cleaved product, it would be useful to show that a specific serine protease inhibitor reduces the appearance of this band at the elevated temperature. While substitution in the proposed active site (D439N) reduces the signal, a specific protease inhibitor would support that the activity in the wild type and disease associated mutant forms have self-cleavage protease activity. It would also be helpful to indicate where the epitope for the antibody is located that can recognize the full-length and cleaved form.

This is a good suggestion. In the revised manuscript, we included new data showing that as expected, the auto-cleavage activity of purified recombinant FAM111A and FAM111B proteins that is visible as the formation of faster migrating bands can be fully suppressed in the presence of the serine protease inhibitor AEBSF (new Figure EV1F,G). Together with the observation that inactivating substitutions in the FAM111A and FAM111B protease domains abolish the formation of these faster migrating bands (Figure 4K; Figure 5L), this firmly demonstrates that these species represent products of FAM111 auto-proteolytic activity. In these experiments we used a FLAG antibody for detection of the recombinant FAM111A and FAM111B proteins (now mentioned in the figure legends), both of which contain an N-terminal FLAG tag, therefore only N-terminal cleavage products can be seen in these blots.

Minor

1. The authors predict the structure of FAM111A and FAM111B by modeling it on the solved crystal structure of the bacterial protease DegS. However, it is not always clearly written in the text that the predicted location of various residues in FAM111A/B are predicted and not solved. For example, the various disease-associated mutant residues are predicted be located outside of the core protease domain.

We have edited the manuscript text accordingly to emphasize that the FAM111A and FAM111B protease domain models are based on predictions from our *in silico* analysis and do not represent experimentally solved structures.

Referee #2:

This manuscript describes FAM111A and FAM111B as proteases that are hyperactivated by disease causing mutations. Their overexpression or hyperactivation yields defects in several biological processes including replication, transcription, and microtubule organization. The data convincingly demonstrates that the disease-causing mutations are gain of function and provide some mechanistic explanations for the biological consequences such as interaction with and

displacement of the RFC protein complex from chromatin. Overall, the data is very high quality and the conclusions are appropriate. I offer the following comments to the authors for their consideration to improve their interesting and well-written manuscript.

1. Figure 1I: Was the level of FAM111A and FAM111B expression similar? How about the mutants? Documentation would be useful in this figure since the authors do compare effects. The authors may also want to comment in the text on why the protease catalytic mutants are often overexpressed at higher levels (presumable selection since they are less toxic to the cells).

As shown in Figure EV5D, the expression levels of wild-type (WT) FAM111A and FAM111B in our inducible cell lines are overall similar. In the revised manuscript, we decided to rearrange the original Figure 1I, so that the cell viability data for all FAM111B cell lines (WT, S628N and S628N/D544N) are now displayed together in a single panel (Figure 5J), which we feel is a more appropriate way to present these results. Expression levels of the GFP-FAM111A alleles in Figure 1I are shown in Figure 2A; we now clearly indicate this in the text and figure legend. Correspondingly, levels of inducibly expressed GFP-FAM111B alleles in the experiment in Figure 5J are shown in Figure 5E, as indicated in the figure legend.

We agree with the referee's suggestion to comment in the text on the notion that the catalytically inactive FAM111A mutant (D439N) is expressed at a higher level than the WT protein (Figure 2A). We believe the reason for this is indeed that expression of inactive FAM111A is not toxic to cells, and we now mention this in the text (page 5).

2. FAM111A was reported to be enriched at replication forks in Wessel et al., Cell Reports 2019 (<https://doi.org/10.1016/j.celrep.2019.08.051>) and its knockdown was reported in that paper to cause hypersensitivity to ATR inhibitors suggesting a function in replication. I recommend adding this paper to the references on page 5 where the Groth Nascent chromatin capture reference is utilized. Interestingly, FAM111B was not identified as a fork-associated protein. While negative data that is difficult to interpret, this is somewhat surprising given the authors documentation that 111A and 111B interact and overexpression of either causes similar problems during replication. The authors may want to comment on this in relation to their observation that only the disease causing 111B mutant proteins generate the same level of replication problems.

As suggested by the referee, we now cite the study by Wessel et al. (Cell Reports 24:3497-3509 (2019)) when mentioning the known association of FAM111A with the replication fork (page 5). In that paper, knockdown of FAM111A was found to reduce cellular sensitivity to ATR inhibitors. While we do not know the reason for this, one interesting possibility is that the potent ability of FAM111A to trigger programmed cell death by apoptosis might be functionally relevant under conditions of severe replication stress. Accordingly, cells lacking FAM111A might be less prone to undergo apoptosis under these conditions and in turn show resistance to killing by ATR inhibitors, a notion that would be interesting to explore in the future.

Consistent with published iPOND datasets, our own iPOND experiments show no detectable association of FAM111B with replication forks, and contrary to FAM111A, FAM111B does not form replication-associated nuclear foci in our hands (Appendix Figure S1A; new Figure EV5A); we now point this out in the text (page 10). This suggests that FAM111A and FAM111B may not engage in stable complex formation at replication forks. In general, however, the precise significance of FAM111A association with the replication fork via its PCNA-binding PIP box remains unclear, given our findings that a FAM111A *PIP mutant that does not interact with PCNA

(Alabert et al., Nature Cell Biol. 16:281-293 (2014)) and does not form replication foci (Appendix Figure S1A) is fully proficient for suppressing DNA replication via its catalytic activity (Figure 2D). Irrespective of the precise underlying mechanism, this helps to explain why deregulated FAM111B protease activity phenocopies the impact of elevated FAM111A proteolytic activity on DNA replication, despite the lack of a recognizable PIP box and stable association with replication forks.

Referee #3:

Elucidating the functions of proteins involved in clinical syndromes is a vital part in understanding how to develop treatments. Hoffmann et al. report both the function of FAM111A and B, as well as the significance of patient mutations in these genes leading to a broad spectrum of disease manifestations. The authors first show that human FAM111A and FAM111B are active proteases in vitro and when FAM111A is overexpressed it substantially lowers cell viability through inhibition of replication and transcription, induction of DNA damage response and activation of apoptosis. They also show that the patient-associated mutations in both proteins result in overactive proteases and they phenocopy the overexpression of FAM111A. To identify a mechanism of the profound effect on replication, the authors show via an interactome analysis as well as various co-IP experiments, that FAM111A interacts with RFC1, and that increased FAM111A protease activity leads to hindered DNA replication by causing RFC1 and PCNA displacement from chromatin.

Overall, this is an excellent manuscript that should be published with minor modifications. The major issue, which the authors are aware of, is that they do not have an actual substrate list for the two proteases and ultimately this is what it would take to understand the plethora of phenotypes associated with the mutations. However, the manuscript has a beautiful set of experimental evidence about the function of the proteases that will be a starting point to fully understand the protease function and the diseases associated with their overactivation.

One issue that should be addressed before publication is the lack of statistical significance indication in any of the dot-plots. That information should be included even if the differences are obvious.

We carefully considered this point, and after consulting with the editor we concluded that it is not appropriate to perform statistical analysis of the representative experiments showing quantitative image-based cytometry (QIBC) analysis of large numbers (typically thousands) of individual cells. These experiments are similar in nature to conventional flow cytometry analyses, for which it is also common practice to depict results as representative experiments. Indeed, other papers reporting QIBC data are also showing representative experiments and consequently not performing statistical analysis of the data (please see Ercilla et al., Cell Reports 30:2416-2429 (2020) and Toledo et al., Cell 155:1088-1103 (2013) for examples). Instead, we added a new supplementary figure (new Appendix Figure S6) in which we show results from independent repeats of most of the QIBC experiments in our study, demonstrating the reproducibility of the observed effects.

Other points:

-The authors identify RFC1, 2, 5 as enriched (Figure 2G). If they do not want to reveal the whole list, can they show some GO analysis of the enriched proteins?

The full list of potential FAM111A-interacting proteins identified in our proteomic analyses (Figure 2G) is provided in Dataset EV1.

-The authors mention that the literature shows FAM111A is important for loading PCNA. Then contradict the literature by saying that with CRIRPR and siRNA of FAM111A, there's no impact on DNA replication, without showing any data on how the KD/KO of FAM111A impacts levels of PCNA. PCNA loading and DNA replication are not interchangeable terms. They should show PCNA levels when FAM111A is KD/KO similarly to Fig 2C if they have it.

This is a very useful suggestion. In the revised manuscript, we included data showing that similar to the lack of effect on overall DNA synthesis rates, cells with targeted knockout of FAM111A, FAM111B or both show no reduction in PCNA chromatin loading in S phase cells (new Appendix Figure S2C,G).

- "These findings suggest that FAM111A inhibits DNA replication by promoting RFC and PCNA displacement from chromatin via its intrinsic protease activity" seems like an overstatement.

Ultimately, to make this conclusion, more work would be needed on replacing the missing RFC and PCNA to the chromatin to see if proper replication could be restored. Overexpressing these proteases likely effects many other proteins as well (besides just RFC and PCNA), ultimately leading to replication defect and increased DNA damage.

We concur with this notion and have rephrased the sentence accordingly, so that it now reads as follows: "These findings show that FAM111A proteolytic activity strongly inhibits DNA replication, involving the displacement of both RFC and PCNA from chromatin." (page 7).

Dear Niels,

Thank you for the submission of your revised manuscript. I have taken over its handling as Martina is currently not in the office. We have now received the enclosed referee reports, and I am happy to say that your manuscript can in principle be accepted, if the last comments by referee 1 can be successfully addressed. Please send us a point by point response to these last comments.

A few more minor changes will also be required:

Please add up to 5 keywords when uploading your manuscript into our online system.

Please add the name "DATASET EV 1" to the file itself, before the legend.

I attach to this email a related manuscript file with comments by our data editors. Please address all comments in the final manuscript.

Best wishes,
Eshter

Referee #1:

This revised manuscript by Hoffmann and colleagues reports on the protein paralogs FAM111A and FAM111B. These two proteins are predicted to contain trypsin-like serine protease domains, have been implicated as viral restriction factors, and present as heterozygote dominant mutants in several poorly characterized diseases. In addition, several reports have implicated FAM111A in chromatin binding, PCNA binding, and DNA replication regulation although most reports have been incomplete. The current report represents a significant advance by reporting that FAM111A and FAM111B can bind to each other, overexpression of FAM111A or FAM111B causes decreased DNA replication and RNA transcription, and the disease associated mutants have increased protease activity with phenotypic behavior similar but stronger to the wild type alleles. Overall this report brings interesting new insights into these proteins that helps to understand the physiologic role of these two enigmatic proteins and their associated diseases.

In general, the authors have responded well to all three reviewers's critiques.

I have a few remaining questions.

Does expression of the cleaved form of FAM111A have activity?

I have several questions related to the *PIP mutant used in several figures. Does this mutant have

wild type activity in all assays shown in the manuscript? For example, the *PIP mutant behaves similar to WT in Figures 2C-F, 2J-K, EV3H reducing levels of EdU incorporation and chromatin bound RFC1 and PCNA but not levels of chromatin bound MCM2.

"Unlike the FAM111A protease domain, mutation of the PIP box that abrogates PCNA binding [3] and localization to PCNA-positive replication foci (Appendix Figure S1A) did not impair the ability of FAM111A to suppress DNA replication and PCNA loading (Figure 2A,C-E)."

Does WT FAM111A localize to PCNA foci? It's not obvious in Figure S1A that either WT or PIP* localize to PCNA foci.

I'm not sure why a statistical test for significant differences can't be applied to the scatter plots in Figure 2 (B, D, E, F, and K), 3 (C, E, G, I)

Referee #3:

I have no more comments. This is an excellent paper on a clearly complex protein.

Point-by-point reply to Referee #1's comments***Does expression of the cleaved form of FAM111A have activity?***

While we do not know the precise location of the auto-cleavage site(s) in FAM111A, we consider it unlikely that the observed cleaved forms retain activity for the following reason: As can be seen in Fig. 4K, the auto-cleaved form of recombinant N-terminally FLAG-tagged human FAM111A (representing an N-terminal fragment as the blot was probed with FLAG antibody), migrates around 62 kDa. Based on the migration of the purified full-length FLAG-FAM111A protein (around 90 kDa), we estimate that cleavage most likely occurs in the region spanning amino acids 420-460. Considering that amino acids 385, 439 and 541 make up the FAM111A protease domain catalytic triad (see Fig. 1B), the auto-cleavage is unlikely to give rise to products that retain protease activity.

I have several questions related to the *PIP mutant used in several figures. Does this mutant have wild type activity in all assays shown in the manuscript? For example, the *PIP mutant behaves similar to WT in Figures 2C-F, 2J-K, EV3H reducing levels of EdU incorporation and chromatin bound RFC1 and PCNA but not levels of chromatin bound MCM2.

In virtually all assays shown in the manuscript, the *PIP mutant behaves essentially like FAM111A WT. To this end, we show in Fig. EV3I that the *PIP mutant suppresses transcription to a similar degree as WT FAM111A. In addition to the impact of the *PIP mutant on DNA replication- and transcription-related readouts that were already shown in the manuscript, we have included additional data showing that induction of the *PIP mutant also triggers apoptosis, albeit with moderately reduced efficiency as compared to FAM111A WT (new Fig. EV3C).

"Unlike the FAM111A protease domain, mutation of the PIP box that abrogates PCNA binding [3] and localization to PCNA-positive replication foci (Appendix Figure S1A) did not impair the ability of FAM111A to suppress DNA replication and PCNA loading (Figure 2A,C-E)."***Does WT FAM111A localize to PCNA foci? It's not obvious in Figure S1A that either WT or PIP* localize to PCNA foci.***

We agree with the referee that the localization of stably expressed GFP-FAM111A WT to PCNA foci is difficult to appreciate in our cell lines (we find that this is more easily seen upon transient overexpression of GFP-FAM111A WT (not shown in the manuscript)), and we therefore re-worded the sentence highlighted by the referee, so that it now reads: "Unlike the FAM111A protease domain, mutation of the PIP box that abrogates PCNA binding [3] did not impair the ability of FAM111A to suppress DNA replication and PCNA loading (Figure 2A,C-E)."

I'm not sure why a statistical test for significant differences can't be applied to the scatter plots in Figure 2 (B, D, E, F, and K), 3 (C, E, G, I)

We do not think it is meaningful to apply statistical testing to the scatter plots showing representative quantitative image-based cytometry (QIBC) experiments, as we believe this would require treating each of the thousands of cells in an individual QIBC experiment as a separate n , which we think is not appropriate as they clearly do not represent independent samples or separate experiments. Therefore, in agreement with the Editors, we did not perform statistical testing of the representative QIBC experiments shown in the manuscript but instead added a separate Appendix Figure (Appendix Figure S6) with repeats of these experiments, showing reproducibility of the results.

Prof. Niels Mailand
NNF Center for Protein Research, University of Copenhagen
Ubiquitin Signaling Group, Protein Signaling Program
Blegdamsvej 3B
Copenhagen DK-2200
Denmark

Dear Niels,

Thank you for the submission of your revised manuscript. I am now very pleased to accept it for publication in the next available issue of EMBO reports. Thank you for your contribution to our journal.

At the end of this email I include important information about how to proceed. Please ensure that you take the time to read the information and complete and return the necessary forms to allow us to publish your manuscript as quickly as possible.

As part of the EMBO publication's Transparent Editorial Process, EMBO reports publishes online a Review Process File to accompany accepted manuscripts. As you are aware, this File will be published in conjunction with your paper and will include the referee reports, your point-by-point response and all pertinent correspondence relating to the manuscript.

If you do NOT want this File to be published, please inform the editorial office within 2 days, if you have not done so already, otherwise the File will be published by default [contact: emboreports@embo.org]. If you do opt out, the Review Process File link will point to the following statement: "No Review Process File is available with this article, as the authors have chosen not to make the review process public in this case."

Should you be planning a Press Release on your article, please get in contact with emboreports@wiley.com as early as possible, in order to coordinate publication and release dates.

Thank you again for your contribution to EMBO reports and congratulations on a successful publication. Please consider us again in the future for your most exciting work.

Kind regards,

Martina

Martina Rembold, PhD
Editor
EMBO reports

THINGS TO DO NOW:

You will receive proofs by e-mail approximately 2-3 weeks after all relevant files have been sent to our Production Office; you should return your corrections within 2 days of receiving the proofs.

Please inform us if there is likely to be any difficulty in reaching you at the above address at that time. Failure to meet our deadlines may result in a delay of publication, or publication without your corrections.

All further communications concerning your paper should quote reference number EMBOR-2020-50662V3 and be addressed to emboreports@wiley.com.

Should you be planning a Press Release on your article, please get in contact with emboreports@wiley.com as early as possible, in order to coordinate publication and release dates.

Corresponding Author Name: Niels Mailand

Journal Submitted to: EMBO REPORTS

Manuscript Number: EMBOR-2020-50662